# The Mediating Role of Healthy Lifestyle Behaviours on the Association between Perceived Stress and Self-Rated Health in People with Non-Communicable Disease

**DOI:** 10.3390/ijerph191912071

**Published:** 2022-09-23

**Authors:** Lena Nordgren, Petra von Heideken Wågert, Anne Söderlund, Maria Elvén

**Affiliations:** 1Centre for Clinical Research Sörmland, Uppsala University, 631 88 Eskilstuna, Sweden; 2Department of Public Health and Caring Sciences, Uppsala University, 751 22 Uppsala, Sweden; 3School of Health, Care and Social Welfare, Mälardalen University, 722 20 Västerås, Sweden

**Keywords:** adults, diet, healthy, exercise, health, health behaviour, healthy lifestyle, mediation analysis, middle-aged, noncommunicable diseases, sedentary behaviour, stress, psychological

## Abstract

Perceived stress can affect people’s lifestyle behaviours and self-rated health. A balanced, healthy lifestyle can alleviate experiences of stress. For clinicians to use evidence-based and theory-based knowledge in health dialogues with people with non-communicable diseases, and in order to develop more effective behavioural counselling, more knowledge is needed. Hence, this study aimed to examine the mediating role of sedentary behaviour, daily physical activity, physical exercise, and dietary habits on the association between perceived stress and self-rated health in people with or without one to four self-reported non-communicable diseases (myocardial infarction, stroke, hypertension, diabetes). The study used a cross-sectional design. Responses from in total 10,583 individuals were collected in 2016 and 2019 by a self-report questionnaire. A series of simple and multiple regression analyses were conducted to examine the mediating effects of healthy lifestyle behaviours on the association between perceived stress and self-rated health. The results show that the investigated healthy lifestyle behaviours partly mediated the association between perceived stress and self-rated health in people with no diagnosis, and in people with one or two diagnoses. It can be concluded that healthy lifestyle behaviours could probably be targeted in relation to the number of noncommunicable diseases that the individuals have.

## 1. Introduction

Stress is a major public health problem at present [1]. It involves the experience of being overwhelmed or unable to handle mental or emotional pressure [2]. Stress affects health directly through autonomic and neuroendocrine reactions, but also indirectly through changes in health behaviours, i.e., unhealthy eating, less exercise or more time spent sedentary [1,3]. Many people are vulnerable to stress but people with non-communicable diseases (NCDs) such as cardiovascular disease (CVD) and diabetes are particular susceptible to stress due to their ill health. In addition, having more than one NCD is common among middle-aged people [4,5]. NCDs also have negative impact on people’s perceived health [4,6]. In addition, as pointed out by Mozaffarian [7], individuals with increased risk of developing NCDs need intensive behavioural counselling, e.g., evidence-based behavioural counselling, health record screening and tracking, referrals to proper services, and quality-of-care standards.

On the contrary, healthy lifestyle behaviours have positive effects on people’s perceived stress, health and wellbeing [8]. Healthy lifestyle behaviour, thus, is emphasized as a central part of evidence-based guidelines worldwide. Accordingly, a change in lifestyle is often recommended for the prevention of complications and control of NCDs [9]. Recommended lifestyle changes usually include reduced sedentary behaviour, increased daily physical activity and/or physical exercise, building healthy dietary habits, and stress management. In spite of that, people with NCDs such as CVD and/or diabetes frequently do not follow the recommendations concerning healthy lifestyle behaviours [10,11,12,13,14]. This is partly caused by a lack of assistance and support regarding how to perform recommended changes to accomplish healthy lifestyle behaviours [15,16]. Since inadequate adherence with recommended health behaviours is a serious obstacle for effective treatment, it is of great importance to identify factors that can be used in individualized clinical practice to support lifestyle changes, as well as to design and implement appropriate interventions.

In several Swedish regions, so-called Health Programmes have been implemented [17]. The Health Programmes aim to empower residents to gain control over and improve their health [17]. The Health Programmes usually include risk factor monitoring. Furthermore, the participants are asked to complete a comprehensive questionnaire that covers socioeconomic and psychosocial conditions, self-reported health, family history of CVD and diabetes, quality of life, and health behaviours [18]. The results of the examination and questionnaire are followed up in a health dialogue with a registered nurse/district nurse who uses Motivational Interviewing [17,19]. Previous studies have shown that the Health Programmes are cost-effective [20], have positive health effects on blood pressure, glucose and smoking [21], and can reduce all-cause and CVD mortality [18,22]. In addition, participants have described that they appreciate the health dialogues since they were perceived as respectful and neutral, and raised an opportunity to reflect on their own lifestyle behaviours, which in turn motivated behavioural change [23].

Counselling interventions need to be both evidence-based and theory-based. According to the Health Belief Model [24,25], five basic conditions must be met for people to have incentives for lifestyle changes. These conditions are that they believe 1. they are susceptible to the disease in question, i.e., they have signs of a disease or they already have a disease; 2. the condition has negative and/or serious consequences for their life situation; 3. a certain treatment regimen or change in behaviour will reduce the threat from the disease; 4. that any negative consequences from the treatment regimen or behavioural change will be outweighed by its positive effects; and 5. they are capable of carrying out the recommended activity. These theoretical points support the assumption that a specific set of evidence-based health recommendations pertinent to a particular group of people would permit the planning of targeted and more effective programmes.

To summarize, it is well-known that perceived stress can affect people’s health, that stress affects people’s health behaviours, and that a balanced, healthy lifestyle *can alleviate experiences of stress*. It is also well-known that people with one or more NCD perceive their health as worse compared to healthy individuals [4]. However, for clinicians to use evidence-based and theory-based knowledge in health dialogues with people with NCDs, and in order to develop more effective behavioural counselling, more knowledge is needed. The complex and interrelated nature between stress, health, and health behaviours raises questions about whether people with formerly known NCDs have the same needs regarding information, education, and support for a change in health behaviour as people without NCDs, or whether the needs differ. Thus, this study aimed to examine whether lifestyle behaviours (sedentary behaviour, daily physical activity, physical exercise, and dietary habits) can mediate the association between perceived stress and self-rated health in people with or without one or more of the following self-reported NCDs: myocardial infarction, stroke, hypertension and/or diabetes (Figure 1).

## 2. Materials and Methods

The study was conducted with a cross-sectional design with data from two different datasets: 2016 and 2019.

### 2.1. Study Context

The Sörmland Health Programme (SHP) was implemented in Region Sörmland, Sweden, in 2014 with inspiration from another similar initiative in Sweden, the Västerbotten Intervention Programme (VIP), that had been running since 1985. The main objective of the programme is to prevent CVD and type 2 diabetes, and to promote healthy lifestyle behaviours [26,27]. The SHP is an initiative where all inhabitants in Region Sörmland are invited when they are 40, 50 and 60 years old to a health dialogue with a trained nurse at a local primary health care centre. The structure of the health dialogue was replicated from the VIP as this was an already well-known and established programme. Besides the health dialogue, the programme also includes a comprehensive self-report questionnaire and risk factor monitoring (for more information see, for example, Norberg et al. [27]). The self-report questionnaire contains questions about socioeconomic status, self-rated health, (life satisfaction, social support, lifestyle (dietary habits, alcohol intake and smoking, sedentary behaviour, physical activity), and history of CVD and diabetes [28].

### 2.2. Sample

The total sample consisted of 10,583 individuals. The 2016 sample included 5837 individuals born in 1956, 1966, or 1976, and the 2019 sample included 4746 individuals born in 1959, 1969, or 1979. In the 2016 sample, 53 per cent of the region’s inhabitants born in 1956, 1966, or 1976, participated in the Health Programme. In the 2019 sample, the corresponding number was 55 per cent.

### 2.3. Data Collection

Data were collected with a self-report questionnaire. The questions were based on the well-known questionnaire that had been in use for several years within the VIP [22,29,30].

#### 2.3.1. Dependent Variable

Self-rated health was used as the dependent variable. It was measured with a single item from the 36-Item Short-Form Survey Instrument (SF-36): *In general, would you say your health is…* The responses were graded on a five-point Likert scale ranging from 1 (very good) to 5 (very poor).

#### 2.3.2. Independent Variable

The independent variable was perceived stress. For both years, stress was defined as a condition of being tense, restless, nervous, worried, or unfocused. However, the two datasets differed regarding the formulation of the question. In 2016, the question read *What level of stress have you experienced in the past month?* The scale was between 1 (low level of stress) and 10 (high level of stress). In 2019, the participants were asked *Have you felt stressed during the last month?* The responses were graded on a four-point Likert scale ranging from 1 (not at all) to 4 (yes, very much). For the purpose of the present study, the responses from 2016 were divided into four groups that corresponded to the response options from 2019, i.e., score 1 to 2 in the 2016 questionnaire were recoded to response option 1 in the 2019 questionnaire, score 3 to 5 were recoded to 2, score 6 to 8 were recoded to 3, and score 9 to 10 were recoded to 4.

#### 2.3.3. Mediator Variables

Mediator variables were sedentary behaviour, daily physical activity, physical exercise, and dietary habits.

Sedentary behaviour was measured with the question: *In general, how many hours do you spend sitting during one day?* The responses were graded on a five-point Likert scale ranging from 1 (0 to 4 h) to 5 (more than 15 h).

Daily physical activity is defined as any movement a person does. It was measured with the question: *How much time do you spend a regular week doing everyday activities, such as walking, cycling or gardening?* Add up all time during a regular week (at least 10 min at a time). The responses were graded on a seven-point Likert scale ranging from 1 (0 min/no time) to 7 (5 h or more).

Physical exercise is by definition planned, structured, repetitive and intentional movement. It was measured with the question: *How much time do you spend a regular week doing physical exercise that causes you to become short of breath, such as running, gymnastics or ball sports?* Add up all time during a regular week. The responses were graded on a six-point Likert scale ranging from 1 (0 min/no time) to 6 (2 h or more).

Dietary habits were measured with six different questions: *How often do you eat/choose vegetables/root crops; fruits/berries; fish/seafood; buns/cakes/chocolates/candy/snacks/soft drinks/lemonade; light products; and unsaturated fats*? The responses about vegetables, and fruit were graded on a four-point Likert scale ranging from 1 (twice a day) to 4 (once a week or less). The responses about fish were graded on a four-point Likert scale ranging from 1 (twice a day) to 4 (once a month or less). Responses about buns/cakes/chocolates/candy/snacks/soft drinks/lemonade ranged from 1 (daily) to 4 (once a week or less). Before the analyses, the responses were inverted. For light products and unsaturated fats, the response options ranged from 1 (always) to 4 (never). Next, the responses were summarized, and a total score was calculated for each participant. The total score ranged from 6 to 24. The total score was recoded into four groups ranging from 1 (healthy) to 4 (unhealthy).

### 2.4. Data Management and Statistical Analyses

The statistical analyses were performed with SPSS v22 (IBM Corp. Released 2013. IBM SPSS Statistics for Windows, Version 22.0. Armonk, NY, USA). The datasets (2016 and 2019) were checked for possible differences regarding demographic data (i.e., sex, age, country of birth, educational level, employment status) with Chi^2^. There were no significant differences between the two datasets regarding sex or age, but there were significant differences regarding country of birth, educational level, and employment status. However, the differences were small and considered not to have any major impact on the results (Table 1).

The variance inflation factor was very close to 1.0, and the tolerance statistics were all well above 0.2, which indicates that there was no collinearity within the data (Field, 2013). Additionally, the bivariate correlations between the independent variable and mediator variables were less than 0.7 throughout the analyses. Therefore, multicollinearity should not have biased the regression analyses [31].

According to Baron and Kenny [32], there are four criteria that need to be met for causal mediation (see also Figure 2):The independent variable must be correlated with the dependent variable.The independent variable must be correlated with the mediator.The mediator must be correlated with the dependent variable when controlled for the independent variable.The effect of the mediator on the dependent variable is controlled. If the independent variable is no longer correlated with the dependent variable, there is a complete mediation. If the correlation between the independent variable and the dependent variable is reduced, there is a partial mediation. If the direct relationship of the independent variable to the dependent variable is less in the fourth regression than in the first, partial mediation is present, which, according to Baron and Kenny [32] is more realistic than perfect mediation.

The mediator analyses were conducted separately for participants who did not report any of the requested diagnoses and for participants who reported one, two, or three to four of the requested diagnoses, i.e., a total of four analyses. Missing data were non-systematic and excluded pairwise.

First, self-rated health (i.e., the dependent variable) was regressed on perceived stress (the independent variable) with simple linear regression. Secondly, sedentary behaviour, daily physical activity, physical exercise, and dietary habits (the mediators) were regressed separately on perceived stress with simple linear regression. Next, in a multiple linear regression analysis, self-rated health was regressed on sedentary behaviour, daily physical activity, physical exercise and dietary habits. Thirdly, perceived stress was controlled for in separate analyses. Fourthly, the effect of the mediation was estimated. For each path, standardized beta coefficients (*β*) and their significance levels were reported [33]. In order to determine if the reduction in the effect of the independent variable was significant even after the mediators were included in the model, an interactive calculation tool was used to conduct the Sobel test [34]. Unstandardized regression coefficients were used in the Sobel tests.

The significant path coefficients between perceived stress, mediators, and self-rated health from the models were used to calculate the strength of the relationship between these variables due to direct, indirect, and total structural effects for all groups [35].

## 3. Results

Demographic data are presented in Table 1. An overview of the distribution of responses in the two different datasets (2016, 2019) and for the total sample is presented in Table 2. Among participants with none of the requested diagnoses 84.0% reported perceived stress at any level during the past month. Corresponding percentages for participants with one diagnose was 86.6%, for two diagnoses 83.3%, for three diagnoses 74.3%, and for all four diagnoses 80.0%. The difference between the groups was not significant *X*^2^ (12, *N* = 9527) = 18.2, *p* = 0.110. Regarding self-rated health, 77.9% of the participants with none of the requested diagnoses reported good or very good health. Corresponding percentages for participants with one diagnose was 64.8%; two diagnoses 46.4%; three diagnoses 27.5%, and for four diagnoses 0.0%. The difference between the groups was significant *X*^2^ (16, *N* = 10,564) = 506.7, *p* = <0.001.

### 3.1. Model Testing and Mediation Model Analysis

#### 3.1.1. No Diagnosis

For participants who did not report any of the requested diagnoses, steps 1 and 2 are presented in Table 3. Perceived stress had a significant direct effect on self-rated health and each of the mediators were significantly associated with perceived stress.

In step 3, the results showed that perceived stress and the mediators together explained 16% of the variance in self-rated health (*Adj R*^2^ = 0.16, *F* = 233.9, *df* = 5, 6233, *p* < 0.001). When perceived stress was controlled for, all mediators except sedentary behaviour were significantly associated with self-rated health (sedentary behaviour *β* = 0.02, *p* = 0.10; daily physical activity *β* = −0.07, *p* < 0.001; physical exercise *β* = −0.22, *p* < 0.001, dietary habits *β* = 0.04, *p* < 0.001), see Figure 3.

Since one mediator (i.e., sedentary behaviour) was not significant in the first model, another multiple regression analysis was conducted with sedentary behaviour excluded. In this analysis (Figure 4), daily physical activity, physical exercise, and dietary habits were significantly associated with self-rated health (daily physical activity *β* = −0.07, *p* < 0.001, physical exercise *β* = −0.22, *p* < 0.001, dietary habits *β* = 0.04, *p* < 0.001). This model was considered the final model for participants who had not reported any of the requested diagnoses. The significant beta coefficient (*β*) for perceived stress in the final step 3 analysis (*β* = 0.28, *p* < 0.001; Figure 4) was smaller than in step 1 (*β* = 0.30, Table 3). Thus, in step 4, it was established that there was a partial mediation of daily physical activity, physical exercise, and dietary habits between perceived stress and self-rated health. The model is presented in Figure 4.

The significant path coefficients from the model in Figure 4 were used to calculate the strength of the relationships between these variables due to direct, indirect and total structural effects (Table 4). The Sobel test showed that the indirect effect of perceived stress on self-rated health was significant via daily physical activity (*z* = −5.64, *p* < 0.001), physical exercise (*z* = −13.8, *p* <0.001), and dietary habits (*z* = 2.38, *p* = 0.02). The relative magnitude of common variance in percent due to the direct effect of perceived stress on self-rated health was 93%. The relative magnitude of the indirect effect of the mediators was 7%.

#### 3.1.2. One Diagnosis

For participants who had reported one of the requested diagnoses, steps 1 and 2 are presented in Table 5. Perceived stress had a significant direct effect on self-rated health, and each of the mediators except for dietary habits were significantly associated with perceived stress.

In step 3, the results showed that perceived stress together with sedentary behaviour, daily physical activity, and physical exercise explained 16% of the variance in self-rated health (*Adj R*^2^ = 0.16, *F* = 131.4, *df* = 4, 2669, *p* < 0.001). When perceived stress was controlled for, all three mediators were significantly associated with self-rated health (sedentary behaviour *β* = 0.05, *p* = 0.009; daily physical activity *β* = −0.07, *p* < 0.001; physical exercise *β* = −0.23, *p* < 0.001). All three mediators were significant through both paths, implying that they were associated with both perceived stress and self-rated health (Figure 5). The standardized beta coefficient (*β*) for perceived stress in step 3 (*β* = 0.28, *p* < 0.001; Figure 5) was smaller than in step 1 (*β* = 0.31, Table 5). Thus, in step 4, it was established that there was a partial mediation of sedentary behaviour, daily physical activity, and physical exercise on the relation between perceived stress and self-rated health.

The significant path coefficients from the model in Figure 5 were used to calculate the strength of the relationships between perceived stress, sedentary behaviour, daily physical activity, physical exercise, and self-rated health due to direct, indirect and total structural effects (Table 4). The Sobel test showed that the indirect effect of perceived stress on self-rated health was significant via sedentary behaviour (*z* = 2.61, *p* = 0.009), daily physical activity (*z* = −3.65, *p* < 0.001), and physical exercise (*z* = −9.54, *p* < 0.001). The relative magnitude of common variance in percent due to direct effect of perceived stress on self-rated health was 90%. The relative magnitude of indirect effects of sedentary behaviour, daily physical activity, and physical exercise was 10% in the group of participants who had reported one of the requested diagnoses.

#### 3.1.3. Two Diagnoses

For participants who had reported two of the requested diagnoses, steps 1 and 2 are presented in Table 6. Perceived stress had a significant direct effect on self-rated health. In turn, physical exercise was significantly associated with perceived stress.

In step 3, perceived stress and physical exercise together explained 12% of the variance in self-rated health (*Adj R*^2^ = 0.12, *F* = 25.4, *df* = 2, 373, *p* < 0.001). When perceived stress was controlled for, physical exercise was significantly associated with self-rated health (*β* = −0.26, *p* < 0.001). Physical exercise was significant through both paths implying that this variable was associated with both perceived stress and self-rated health (Figure 6).

The standardized beta coefficient (*β*) for perceived stress in step 3 (*β* = 0.17, *p* = 0.001, Figure 6) was smaller than in step 1 (*β* = 0.23, Table 6). Thus, in step 4, it was established that there was a partial mediation of physical exercise between perceived stress and self-rated health. The significant path coefficients between perceived stress, physical exercise and self-rated health from the model in Figure 6 were used to calculate the strength of the relationship between these variables due to direct, indirect and total structural effects (Table 4). The Sobel test showed that the indirect effect of perceived stress on self-rated health was significant via the mediator physical exercise (*z* = −2.90, *p* = 0.004). The relative magnitude of common variance in percent due to direct effect of perceived stress on self-rated health was 74%. The relative magnitude of indirect effect of physical exercise was 26% in the group of participants who had reported two of the requested diagnoses.

#### 3.1.4. Three or Four Diagnoses

For participants who had reported three or four of the requested diagnoses, steps 1 and 2 are presented in Table 7 perceived stress was found to have a significant direct effect on self-rated health (*p* = 0.013). However, none of the mediators were significantly associated with perceived stress. Therefore, steps 3 and 4 were not conducted. Subsequently, the final model for participants who reported three to four diagnoses showed that only perceived stress was associated with self-rated health.

## 4. Discussion

The novelty of this study does not lay in the non-surprising insight that there is a positive impact of healthy lifestyle behaviours on the association between perceived stress and self-rated health. Instead, the novelty lays in the understanding that healthy lifestyle behaviours beneficially can be specified and targeted in relation to the number of NCDs that individuals may have. Hence, in line with previous studies (for example [36,37,38]), the current study showed that perceived stress was significantly associated with self-rated health. The results also showed that for people with none of the requested diagnoses (i.e., hypertension, diabetes, stroke or myocardial infarction), daily physical activity, physical exercise, and healthy dietary habits could reduce the impact of perceived stress on self-rated health. Furthermore, for people with one of the requested diagnoses, more daily physical activity and physical exercise, as well as less sedentary behaviour could decrease the effects of perceived stress on self-rated health. However, the results also showed that for participants with two of the requested diagnoses the relationship between perceived stress and self-rated health was mediated in a positive direction by physical exercise only. One likely explanation for the results lies in the previously well-known relation between physical activity/exercise, stress, and health problems (see for example [39,40]). Exercise reduces stress directly and has also psychological effects. Thus, exercise has a more important role in stress management than for example eating habits and sedentary behaviour. In the current study, there were no significant differences regarding perceived stress among participants irrespective the number of diagnoses. However, participants with one or more diagnoses rated their health as significantly worse than participants with no diagnose. In light of the present findings this could possibly explain the findings about the mediating effect of in particular physical exercise and that the importance increased in relation to the number of diagnoses.

The findings are in line with recent research by for example Edholm et al. [41], who studied physical functioning in older women. They found that physical exercise rather than sedentary behaviour was related to physical function in older age. They also found that older women who had been engaged in physical exercise earlier in life (i.e., between 50 and 65 years of age) were more likely to maintain their physical function when they got older. Based on the current findings, then, it seems appropriate to suggest that health-promoting activities, including support for physical exercise, would above all be relevant for targeted health promoting activities for people in ages 40 to 60 years with one or more NCDs in, for example, health dialogues. In addition, for people with none of the requested diagnoses or with only one of the diagnoses, targeting any health-promoting activity could be effective since people are more likely to make lifestyle changes if they have an incentive or motive to do so [24,25]. That is, if people can choose an activity that they believe will be beneficial for their health situation, they are more likely to actually make a behavioural change. Since the examined health behaviours mediate perceived stress, this provides more choices for people without or with only one diagnosis.

According to the Health Belief Model, it is not enough to only educate and inform about appropriate health behaviours [25]. In order to succeed, i.e., to motivate people to make lifestyle changes, it is also necessary to include activities that can enhance the patients’ self-efficacy to change health behaviour [25]. To motivate patients to start, e.g., exercising, the healthcare provider together with the patient must find a realistic way of realizing the plan. The change in behaviour needs to be gradually introduced, so that the initial tasks or short-term goals are perceived as easy and possible to perform before the difficulty or activity level increases [25].

Rosenstock [24] has suggested three principles that can support patients’ motivation for behavioural change. These principles include strategies directed toward the healthcare system, the healthcare provider, and/or the patient. To modify the healthcare system, one strategy would be to simplify the regimen, example by breaking the treatment plan, or regarding the results of this study, an exercise plan, into a graduated sequence of steps of increasing difficulty. Regarding influencing the healthcare provider, adherence could be enhanced by being attentive to the patients’ educational needs. The previously mentioned health dialogues in the Swedish health programmes are based om motivational interviewing [19], which is in line with the strategy suggested by Rosenstock [24]. The third strategy, to influence the patient, could aim to increase the patients’ self-efficacy for physical activity. Short-term goals, contracts and social support are suggested ways to reinforce adherence [24]. Today, in this contemporary digital world, persuasive technology seems to be desirous to many participants in health programmes [42,43,44]. Thus, there is a need to develop interventions that include persuasive information and communication technology solutions in the context of health promotion and health programmes.

### Methodological Considerations

Obviously, a study such as this holds several methodological considerations. The cross-sectional design of the study, which does not allow any causal conclusions, can be a limitation. One could reason that age would have an impact on number of diagnoses, thus implying a certain causality of the results. However, age did not emerge as a mediator in our analyses; thus, no conclusions can be drawn about causal relationships. Further, there is the question of generalizability. In official statistics from the Public Health Agency of Sweden, 73% of the general Swedish population aged 45 to 64 rated their health as good in 2021 [45], which equals the results from the present study. There were slightly more women than men in the present study (women 53%, men 46%) than in the general Swedish population (50/50) [46]. Furthermore, 77% of the participants in the present study were born in Sweden, compared to 80% in the general Swedish population [46]. Then again, the prevalence of hypertension in the general Swedish population has been estimated to approximately 27% [47]. The corresponding figure for diabetes is 5% [48]. Moreover, in the general Swedish population it is estimated that approximately half of the patients with diabetes type 2 also have hypertension [47]. These figures correspond to the population in the present study (hypertension 29.8%, diabetes 3.9%, both hypertension and diabetes 2.5%). Hence, in light of the present study, it can be assumed that the generalizability of the current findings to a general Swedish population is tolerable.

There were no tests for reliability or validity of the questionnaire in whole. However, the question measuring self-rated health is included in the well-established, reliable and valid SF-36 [49] and has been used as single question in several studies (for example [50,51]). The other questions stem from other established questionnaires such as IPAQ-SF [52], but were slightly modified to fit the context. The VIP has been continuing since the 1980s and the programme as well as the questionnaire is well-established. The experience-based practice of the questionnaire demonstrates that users agree on its relevance and appropriateness; thus, its face validity is supported. Then again, the question about perceived stress differed between the two datasets (see page 4). Both questions have been used for several years within the current health programme as well as in other health programmes. In addition, the question in the 2016 questionnaire is widely used by for example the American Psychological Association (see https://www.apa.org/news/press/releases/stress/index accessed on 15 August 2022). The question in the 2019 questionnaire stem from the “Perceived Stress Scale” which is also a validated and well-known instrument.

Other limitations in this study are that neither body mass index (BMI), alcohol, nor tobacco use were included in the analyses. All these factors are potentially very important regarding health behaviours, which asks for further investigation.

For participants with three or four diagnoses, there were no mediating effects of any of the tested mediators. This is possibly due to the small number of participants with three or four diagnoses and maybe also because their general condition is disturbed due to the numerous diagnoses which causes stress in itself.

## 5. Conclusions

Perceived stress was related to health independently to the number of diagnoses, and for participants with no diagnoses, daily physical activity, physical exercise, and healthy dietary habits could have a positive impact on stress, thus promoting health and possibly preventing ill-health. For people with one diagnosis, behaviours including being more physically active and exercising and less sedentary behaviour could possibly decrease the effects of perceived stress on their health. Additionally, for participants with two diagnoses, the relationship between stress and health was only mediated by physical exercise. Within the scope of health dialogues with people with NCDs, healthy lifestyle behaviours could possibly be targeted in relation to the number of diseases that the individuals have. However, the limitation of low frequency of participants with two diagnoses must be considered when drawing conclusions of that part of our study.

## Figures and Tables

**Figure 1 ijerph-19-12071-f001:**
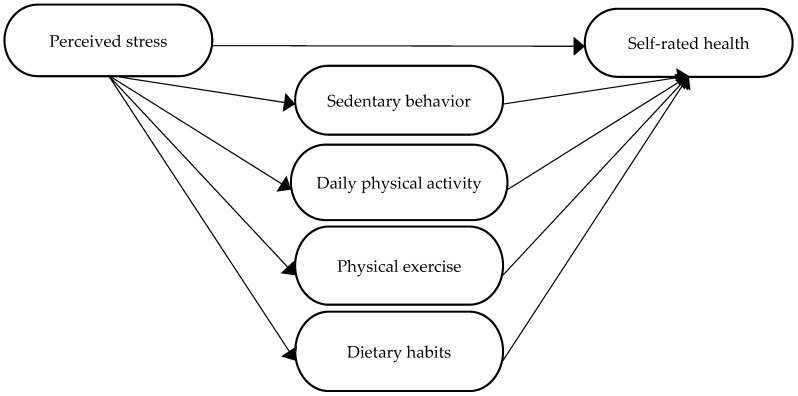
The hypothesized mediation model with four parallel mediators.

**Figure 2 ijerph-19-12071-f002:**
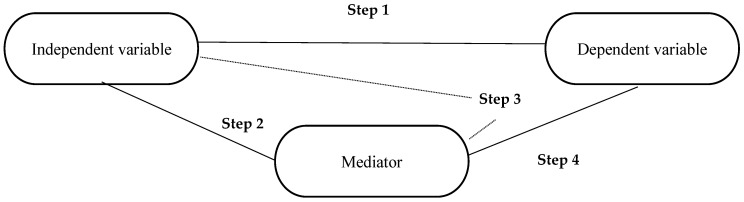
Mediator analysis according to Baron and Kenny [32].

**Figure 3 ijerph-19-12071-f003:**
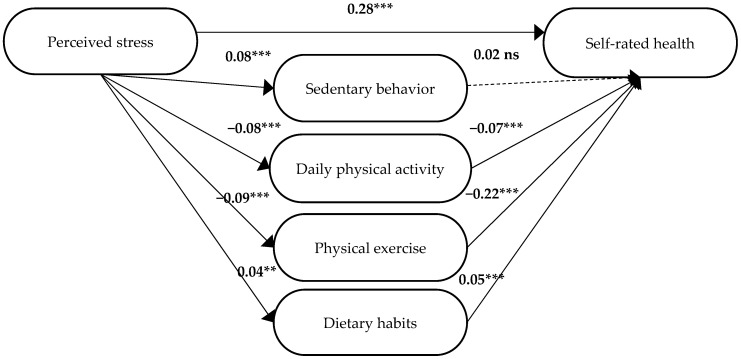
The first model for participants who did not report any of the requested diagnoses. The standardized beta coefficients (*β*) are included for the different paths between the independent variable, i.e., perceived stress, the mediators, i.e., sedentary behaviour, daily physical activity, physical exercise, and dietary habits, and the dependent variable, i.e., self-rated health. ** *p* ≤ 0.01, *** *p* ≤ 0.001.

**Figure 4 ijerph-19-12071-f004:**
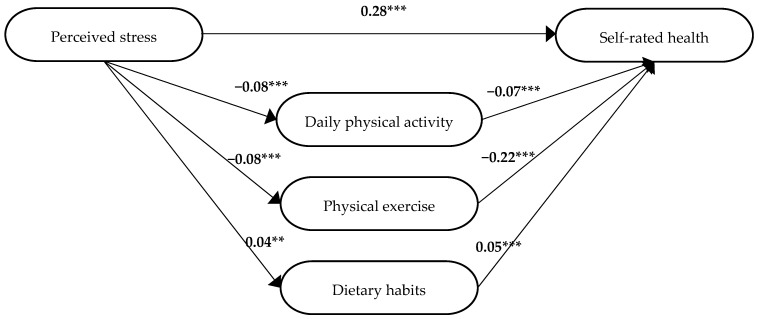
The final model for participants who did not report any of the requested diagnoses. The standardized beta coefficients (*β*) are included for the paths between the independent variable, i.e., perceived stress, the mediators, i.e., daily physical activity, physical exercise, and dietary habits, and the dependent variable, i.e., self-rated health. ** *p* ≤ 0.01, *** *p* ≤ 0.001.

**Figure 5 ijerph-19-12071-f005:**
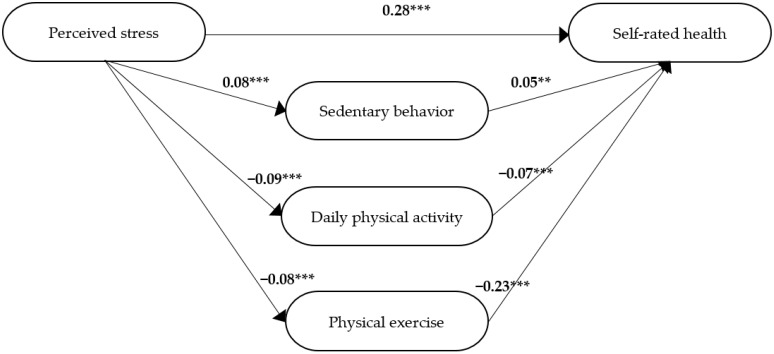
The final model for participants who reported having one of the requested diagnoses. The standardized beta coefficients (*β*) for the different paths between the independent variable, i.e., perceived stress, the mediators, i.e., sedentary behaviour, daily physical activity, and physical exercise, and the dependent variable, i.e., self-rated health. ** *p* ≤ 0.01, *** *p* ≤ 0.001.

**Figure 6 ijerph-19-12071-f006:**
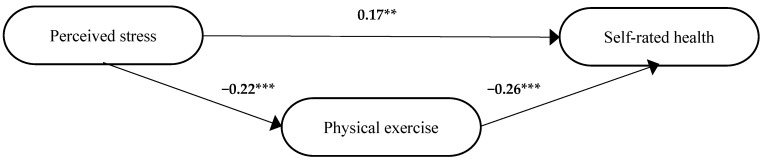
The final model for participants who reported two of the requested diagnoses. The standardized beta coefficients (*β*) for the different paths between the independent variable, i.e., perceived stress, the mediator, i.e., physical exercise, and the dependent variable, i.e., self-rated health. ** *p* ≤0.01, *** *p* ≤ 0.001.

**Table 1 ijerph-19-12071-t001:** Demographic data. Frequencies for the total sample (N = 10,583) and for sub-samples (2016 n = 5837; 2019 n = 4746). Also presented are *p*-values for Chi^2^ tests that were used to check for differences between the two different samples from 2016 and 2019. Missing data are not reported.

Variables	Total; n (%)	2016; n (%)	2019; n (%)	*p.*
Sex				0.162
-male	4868 (46.0)	2695 (46.2)	2173 (45.8)	
-female	5593 (52.8)	3020 (51.7)	2573 (54.2)	
Age				0.472
-40	2900 (27.4)	1574 (27.0)	1326 (27.9)	
-50	3770 (35.6)	2091 (35.8)	1679 (35.4)	
-60	3800 (35.9)	2059 (35.3)	1741 (36.7)	
Country of birth				0.002
-Sweden	8184 (77.3)	4559 (78.1)	3625 (76.4)	
-Nordic countries except Sweden	441 (4.2)	237 (4.1)	204 (4.3)	
-Europe except Nordic countries	400 (3.8)	207 (3.5)	193 (4.1)	
-outside Europe	1440 (13.6)	728 (12.5)	712 (15.0)	
Education level				0.004
-elementary school	1665 (15.7)	946 (16.2)	719 (15.1)	
-upper secondary school	5149 (48.7)	2859 (49.0)	2290 (48.3)	
-higher education	3579 (33.8)	1880 (32.2)	1699 (35.8)	
Employment status				0.017
-employed	7418 (70.1)	4023 (68.9)	3395 (71.5)	
-self-employed	946 (8.9)	545 (9.3)	401 (8.4)	
-unemployed	499 (4.7)	300 (5.1)	199 (4.2)	
-sickness benefit/sickness or activity compensation	565 (5.3)	329 (5.6)	236 (5.0)	
-other	1155 (10.9)	640 (11.0)	515 (10.9)	

**Table 2 ijerph-19-12071-t002:** Distribution of responses in the two data sets (2016, 2019) and in the total sample.

Variables	2016			2019			Total		
	All	Male	Female	All	Male	Female	All	Male	Female
	n (%)								
Self-rated health									
-very good/good	4252 (73.1)	2030 (75.4)	2144 (71.2)	3431 (72.3)	1631 (75.1)	1800 (70.0)	7683 (72.7)	3661 (75.2)	3944 (70.5)
-fair/poor/very poor	1566 (26.9)	662 (24.6)	867 (28.7)	1315 (27.7)	542 (24.9)	773 (30.0)	2880 (27.3)	1204 (24.7)	1640 (29.3)
Felt stress									
-not at all	647 (13.5)	353 (16.0)	274 (11.0)	811 (17.2)	455 (21.0)	356 (13.9)	1458 (15.3)	808 (16.6)	630 (11.3)
-yes, some/yes, pretty much/yes, a lot	4156 (86.5)	1851 (68.7)	2228 (73.8)	3913 (82.8)	1708 (78.6)	2205 (85.7)	8069 (84.7)	3559 (73.1)	4433 (79.3)
Sedentary behaviour									
-≤7 h per day	4096 (70.7)	1801 (66.8)	2212 (73.2)	3590 (75.8)	1600 (73.6)	1990 (77.3)	7686 (73.0)	3401 (69.9)	4202 (75.1)
->7 h per day	1694 (29.3)	876 (32.5)	782 (25.9)	1149 (24.2)	570 (26.2)	579 (22.5)	2844 (23.0)	1446 (29.7)	1361 (24.3)
Daily physical activity									
-<150 min per week	3615 (62.2)	1717 (63.7)	1821 (60.3)	2897 (61.2)	1397 (64.3)	1500 (58.3)	6512 (61.8)	3114 (64.0)	3321 (59.4)
-≥150 min per week	2193 (37.8)	970 (36.0)	1182 (39.1)	1838 (38.8)	771 (35.5)	1067 (41.5)	4031 (38.2)	1741 (35.8)	2249 (40.2)
Physical exercise									
-<1 h per week	3636 (62.7)	1692 (62.8)	1869 (61.9)	2822 (59.6)	1275 (58.7)	1547 (60.1)	6458 (61.3)	2967 (60.9)	3416 (61.1)
-≥1 h per week	2166 (37.3)	997 (37.0)	1124 (37.2)	1911 (40.4)	895 (41.2)	1016 (39.5)	4077 (38.7)	1892 (38.9)	2140 (38.3)
Dietary habits									
-very healthy/healthy	1722 (30.5)	580 (21.5)	1103 (36.5)	1300 (27.7)	419 (19.3)	881 (34.2)	3022 (29.2)	999 (20.5)	1984 (35.5)
-somewhat unhealthy/unhealthy	3921 (69.5)	2032 (75.4)	1816 (60.1)	3390 (72.3)	1727 (79.5)	1663 (64.6)	7311 (70.8)	3759 (77.2)	3479 (62.2)
Diagnosis									
-hypertension	1703 (29.2)	822 (30.5)	844 (27.9)	1449 (30.5)	714 (32.9)	735 (28.6)	3152 (29.8)	1536 (31.6)	1579 (28.2)
-diabetes	218 (3.7)	132 (4.9)	84 (2.8)	192 (4.0)	107 (4.9)	85 (3.3)	410 (3.9)	239 (4.9)	169 (3.0)
-myocardial infarction	142 (2.4)	90 (3.3)	45 (1.5)	76 (1.6)	54 (2.5)	22 (0.9)	218 (2.1)	144 (3.0)	67 (1.2)
-stroke	98 (1.7)	52 (1.9)	44 (1.5)	91 (1.9)	50 (2.3)	41 (1.6)	189 (1.8)	102 (2.1)	85 (1.5)
Number of diagnoses									
-no diagnosis	3958 (67.8)	1775 (65.9)	2101 (69.6)	3170 (66.8)	1400 (64.4)	1770 (68.8)	7128 (67.4)	3175 (65.2)	3871 (69.2)
-one diagnosis	1625 (27.8)	765 (28.4)	827 (27.4)	1366 (28.8)	637 (29.3)	729 (28.3)	2991 (28.3)	1402 (28.8)	1556 (27.8)
-two diagnoses	227 (3.9)	135 (5.0)	86 (2.8)	192 (4.0)	122 (5.6)	70 (2.7)	419 (4.0)	257 (5.3)	156 (2.8)
-three diagnoses	26 (0.4)	19 (0.7)	6 (0.2)	14 (0.3)	12 (0.6)	2 (0.1)	40 (0.4)	31 (0.6)	8 (0.1)
-four diagnoses	1 (0.02)	1 (0.04)	-	4 (0.1)	2 (0.1)	2 (0.1)	5 (0.05)	3 (0.1)	2 (0.04)

**Table 3 ijerph-19-12071-t003:** Simple regression analyses for participants who did not report any of the requested diagnoses (steps 1 and 2).

Variables	*Adj R* ^2^	*F*-Value	*p*-Value	*df* 1, *df* 2	*β*	*p* (for *β*)
Step 1						
Stress—self-rated health	0.09	642.5	<0.001	1, 6394	0.30	<0.001
Step 2						
Stress—sedentary behaviour	0.01	30.5	<0.001	1, 6379	0.07	<0.001
Stress—daily physical activity	0.01	46.2	<0.001	1, 6386	−0.08	<0.001
Stress—physical exercise	0.01	46.5	<0.001	1, 6379	−0.08	<0.001
Stress—dietary habits	0.001	6.1	0.013	1, 6283	0.03	0.013

**Table 4 ijerph-19-12071-t004:** The strength of the relationships between variables due to direct, indirect and total structural effects of perceived stress, sedentary behaviour, daily physical activity, physical exercise, and dietary habits on self-rated health.

Effect	No Diagnosis (n = 7128)	One Diagnosis (n = 2991)	Two Diagnoses (n = 419)
Direct effect (perceived stress)	0.24	0.28	0.17
Indirect effect	0.02 *	0.03 **	0.06 ***
Total effect (direct + indirect)	0.26	0.31	0.23

* daily physical activity, physical exercise, dietary habits ** sedentary behaviour, daily physical activity, physical exercise *** physical exercise.

**Table 5 ijerph-19-12071-t005:** Simple regression analyses for participants who reported one of the requested diagnoses (steps 1 and 2).

Variables	*Adj R* ^2^	*F*-Value	*p*-Value	*df* 1, *df* 2	*β*	*p* (for *β*)
Step 1						
Stress—self-rated health	0.10	295.5	<0.001	1, 2695	0.31	<0.001
Step 2						
Stress—sedentary behaviour	0.01	16.4	<0.001	1, 2692	0.08	<0.001
Stress—daily physical activity	0.01	22.2	<0.001	1, 2695	−0.09	<0.001
Stress—physical exercise	0.01	16.1	<0.001	1, 2695	−0.08	<0.001
Stress—dietary habits	0.0002	0.4	0.517	1, 2648	0.01	0.517

**Table 6 ijerph-19-12071-t006:** Simple regression analyses for participants who reported two of the requested diagnoses (steps 1 and 2).

Variables	*Adj R* ^2^	*F*-Value	*p*-Value	*df* 1, *df* 2	*β*	*p* (for *β*)
Step 1						
Stress—self-rated health	0.51	21.38	<0.001	1, 375	0.23	<0.001
Step 2						
Stress—sedentary behaviour	0.01	1.78	0.182	1, 372	0.07	0.182
Stress—daily physical activity	0.01	0.96	0.328	1, 375	−0.05	0.328
Stress—physical exercise	0.05	19.74	<0.001	1, 375	−0.22	<0.001
Stress—dietary habits	−0.003	0.01	0.919	1, 363	−0.005	0.919

**Table 7 ijerph-19-12071-t007:** Simple regression analyses; participants with three or four of the requested diagnoses (steps 1 and 2).

Variables	*Adj R* ^2^	*F*-Value	*p* Value	*df* 1, *df* 2	*β*	*P* (for *β*)
Step 1						
Stress—self-rated health	0.13	6.72	0.013	1, 38	0.39	0.013
Step 2						
Stress—sedentary behaviour	0.04	2.39	0.131	1, 37	−0.25	0.131
Stress—daily physical activity	−0.03	0.0005	0.983	1, 37	−0.01	0.983
Stress—physical exercise	−0.01	0.50	0.483	1, 37	−0.12	0.483
Stress—dietary habits	−0.02	0.27	0.609	1, 37	0.08	0.609

## Data Availability

The data presented in this study are openly available in FigShare at doi:10.6084/m9.figshare.20513874.

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
