# Peer review of "The Mediating Role of Healthy Lifestyle Behaviours on the Association between Perceived Stress and Self-Rated Health in People with Non-Communicable Disease"

_ijerph, 2022, doi:10.3390/ijerph191912071_

Round 1
Reviewer 1 Report
The manuscript entitled “The mediating role of healthy lifestyle behaviours on the association between perceived stress and self-rated health in people with non-communicable disease” is the study with an aim to examine whether lifestyle behaviours can affect association between perceived stress and health (self-rated) among healthy subjects and those with one of more non-communicable diseases.
The title of the paper is very promising, but at the end the conclusion is a bit far-fetched considering the presented results.
The results show that healthy lifestyle behaviours have an effect (on the relationship between stress and self-rated health) only within the healthy population, and among those with diagnosis of NCDs only physical activity from all observed elements have an effect in population on those with one (but not with two or more) NCD diagnosis.
Is there a possibility that effect of healthy lifestyle is related to the body mass of the subjects?
Body mass, or BMI, can be a very important factor in this case, but you didn't mention it (examine the impact).
One general question is why you didn’t include people/patients with autoimmune diseases such as Hashimoto's thyroiditis for which is known to be exacerbated by stress?
Abstract – maybe to say total responses (row) because it is not clear was there 10583 responses each year or total
Keywords - excessive number of words, in instructions for authors is recommended 3-10 words
Materials and Methods
- Sample – here is not clear enough whether it is a repeated questionnaire on the same sample of subjects
- DH (line 164-166) - how is defined the distribution to food groups? E.g. putting together lemonade with cakes, candy, chocolate,… is maybe to wide; light products – from all food groups?
- - line 190&193 – reference not in in accordance with the instructions (same in the line 204)
- - write the part line194-205 more clearly
Results
- what is the percentage of high educated people within the Swedish population? Is it possible that in the 2 years of research that there were no one subject with education higher than “undergraduate level”
- - result 27.3% of fair to very poor SHR was in average for both year or? (why didn’t you provide separate data for each year)
- According to the results diet could have only impact among healthy people? Is there explanation for that? Is it possible that diet change toward healthier could cause frustration and higher level of stress?
- - line 459-461 – and maybe their general condition due to the numerous diagnosis is disturbed and causes stress in itself
Discussion - the discussion part is a little stretched in the desired direction, it could have been done better, more in acordance with obtained results
Conclusion
For this kind of conclusion you have very poor distribution of respondents e.g. only 4,45% of subject with 2 or more diagnosis
Reviewer 2 Report
Interesting paper, and it's been written in a linear way, the paper has looked at the novel angel and elaborated the concept well. Consider addressing the reviewer's concerns. Some drawbacks include the fact that stats need to be transparent with clear ideas re each step.
Introduction
Why are “Daily physical activity” and “Physical exercise” defined separately in the model? Both are physical activity variables, what is the difference in their meaning?
Methods
Psychological problems, such as stress, require professional scale evaluation. The evaluation of stress is too simple, how to ensure the accuracy?
It is not clear how the independent variable in 2016 was divided into four groups that corresponded to the response options from 2019.
From statistical analyses, I understand Baron and Kenny's mediation model was used. This is however outdated, and it would be better to use MacKinnon's product-of-coefficients method, or causal mediation analysis based on the Potential Outcomes Framework. Also make clear whether covariates were included in the mediation analysis.
Results
There are too many abbreviations in the text, which can easily cause difficulty in reading
Discussion
How to explain the differences in the mediation effect of lifestyle on the number of non-communicable diseases?
Reviewer 3 Report
I consider that the subject of study, the mediating role of healthy lifestyle behaviours on the association between perceived stress and self-rated health in people with non-communicable disease, is very interesting. The paper fits the scope of the journal Int. J. Environ. Res. Public Health. The study is correctly designed. Abstract is too long - it should be a total of about 200 words maximum. Cited references are mostly recent publications and relevant. Complete all the references in accordance with the Instructions for the authors.
I suggest this paper to be published in IJERPH after suitable revision.
Round 2
Reviewer 1 Report
I would like to thank the authors for their efforts to improve the manuscript.
Some rearrangements have been made to the text and some things within the manuscript are now somewhat clearer.
The importance of researching the impact of stress on human health today is unquestionable, and to that extent this work is of great importance.
But, here are still missing some important variables (factors that could have great impact) such as BMI or any anthropometric parameter that are very much related to diet as well as to chronic non-communicable diseases. Other thing that is important to point out is that such important results/conclusions cannot be given on the basis of a such small number of respondents (e.g. 4.5% of the respondents have two ore more diagnosis, and only 0.4% of them have three or more).
Reviewer 2 Report
This is an interesting study, but the independent and dependent variables collected by the authors make me wonder if they reflect the real situation.
Too many influencing factors will affect the results of the mediation analysis, such as the subject's weight and income, which will affect the subject's behavior, and the authors did not include it in the analysis.
In addition, the authors analyzed separately according to the number of diagnoses, but still did not give a reasonable interpretation in the discussion.
